# Strain Relaxation Effect on the Peak Wavelength of Blue InGaN/GaN Multi-Quantum Well Micro-LEDs

Chaoqiang Zhang [1], Ke Gao [1], Fei Wang [1], Zhiming Chen [1], Philip Shields [2], Sean Lee [3], Yanqin Wang [3], Dongyan Zhang [3], Hongwei Liu [1,*] and Pingjuan Niu [1,*]

1 Tianjin Key Laboratory of Optoelectronic Detection Technology and System, School of Electronics and Information Engineering, Tiangong University, Tianjin 300387, China; 2031070895@tiangong.edu.cn (C.Z.); 1931075520@tiangong.edu.cn (K.G.); 2010940523@tiangong.edu.cn (F.W.); 1910940502@tiangong.edu.cn (Z.C.)
2 Department of Electronic and Electrical Engineering, University of Bath, Bath BA2 7AY, UK; p.shields@bath.ac.uk
3 Sanan Optoelectronics Co., Ltd., Xiamen 361009, China; celee@sanan-e.com (S.L.); tj-wangyanqin@sanan-e.com (Y.W.); dyzhang2012@sanan-e.com (D.Z.)
* Correspondence: liuhongwei@tiangong.edu.cn (H.L.); niupingjuan@tiangong.edu.cn (P.N.)

**Abstract:** In this paper, the edge strain relaxation of InGaN/GaN MQW micro-pillars is studied. Micro-pillar arrays with a diameter of 3–20 μm were prepared on a blue GaN LED wafer by inductively coupled plasma (ICP) etching. The peak wavelength shift caused by edge strain relaxation was tested using micro-LED pillar array room temperature photoluminescence (PL) spectrum measurements. The results show that there is a nearly 3 nm peak wavelength shift between the micro-pillar arrays, caused by a high range of the strain relaxation region in the small size LED pillar. Furthermore, a 19 μm micro-LED pillar's Raman spectrum was employed to observe the pillar strain relaxation. It was found that the Raman $E_2^H$ mode at the edge of the micro-LED pillar moved to high frequency, which verified an edge strain relaxation of = 0.1%. Then, the exact strain and peak wavelength distribution of the InGaN quantum wells were simulated by the finite element method, which provides effective verification of our PL and Raman strain relaxation analysis. The results and methods in this paper provide good references for the design and analysis of small-size micro-LED devices.

**Keywords:** InGaN/GaN multiple quantum well (MQW); strain relaxation; micro-LED arrays; photoluminescence (PL); Raman shift

## 1. Introduction

When the size of an LED chip is reduced to tens of microns or even a few microns, it is called a micro-LED chip. Because the micro-display is based on red, green, and blue (RGB) light, micro-LED chips have a high resolution, high brightness, long life, high response speed, and low power consumption. Micro-LEDs have important applications in high-resolution displays, augmented reality, high-speed visible light communication, micro-projectors and other fields [1–3]. Therefore, micro-LED research has been highly valued by researchers in academia and industry all over the world.

In order to improve the luminous efficiency, a multiple quantum well (MQW) structure generally is adopted as the active layer in the micro-LED [4–6]. However, the difference in the lattice constants of the two materials in the quantum well will have a certain impact on the performance of the device. Using the blue LED as an example, stacked InGaN/GaN layers are used to fabricate multiple quantum wells. Due to the ~11% lattice mismatch between InN and GaN when the crystal grows along the *c*-axis, the InGaN/GaN multiple quantum well (MQW) suffers from epitaxial strain and a strong piezoelectric field [7], which leads to a quantum-confined Stark effect (QCSE) that further limits the internal quantum efficiency of InGaN/GaN MQW LEDs [8,9]. In addition, recent studies have indicated that the strain giving rise to the QCSE may be fully or partly relaxed at the boundary of micro-

or nano-scale GaN pillars, stimulating many previous studies on the stress distributions on such samples. For example, Y.Kawakami et al. compared the radiation recombination rate of the quantum well of the micro-column structure at the edge and central regions of the micro-column in a time-resolved spectroscopy test, and the radiation recombination rate of the strain relaxation emission zone was higher than that of the strain zone, verifying the existence of the edge stress release phenomenon [10]. The high-resolution CL test of E.Y.Xie et al. performed a full scan of the sample and observed that there was a difference in the radiation wavelength at the boundary and center of the cylindrical sample [11]. The low photoluminescence measured by Peichen Yu from the embedded InGaN/GaN MQW shows a blueshift energy of 68 meV [12]. The above experiments are conducted with ultra-high-resolution CL and PL spectroscopy, which takes a long time and requires equipment with a very high spatial resolution. Moreover, these studies are all tested on a single MQW pillar and there are no reports studying the edge stress release phenomenon through information obtained from a large number of samples, while the micro-LED applications are generally based on the form of the arrays.

In this paper, several micro-LED arrays of different pillar diameters are fabricated to observe the strain relaxation effect on InGaN/GaN MQWS wavelength modulation. The PL spectra before and after the etching of the micro-LED pillar arrays are used to characterize the MQW edge stress release effect on the arrays' radiation.

Confocal Raman tests are also performed on the single micro-LED pillars to demonstrate the release of lattice mismatch stress on the quantum well sidewalls. Finally, an MQW solid mechanics finite element method (FEM) simulation is used to verify the above analysis.

## 2. Experiments

### 2.1. Epitaxial Growth and InGaN/GaN Micro-LED Arrays Fabrication

The InGaN/GaN LED wafer was grown on a 4-inch c-plane sapphire substrate by metal–organic chemical vapor deposition. The epitaxy layer is illustrated in Figure 1: a 3 μm undoped GaN buffer layer; a 1.5 μm n-type GaN layer; twelve periods 3 nm MQWs separated by 12 nm GaN barrier layers; and a 0.2 μm p-type GaN top layer.

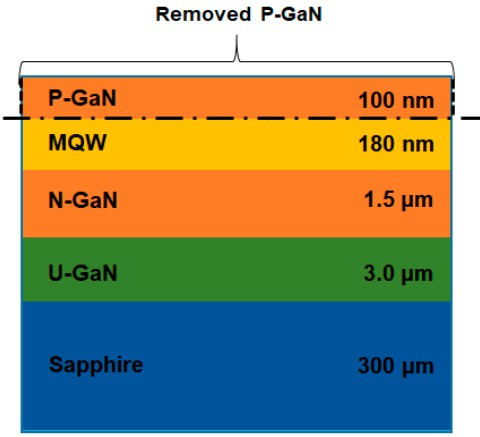

**Figure 1.** Schematic of GaN blue LED epitaxial wafer layers.

In order to study strain relaxation of InGaN/GaN MQWs, the top p-GaN layer thickness was reduced to 100 nm by inductively coupled plasma (ICP) etching. Then, we divided a 4-inch wafer into 18 areas of the same size with area sizes of 12.24 mm × 19.38 mm. We etched an equal number of micro-pillars of different sizes in each area and the etching period was 30 μm (row: 12,240 μm /30 μm = 408, column: 19,380 μm/30 μm = 646). The diameters of the etched micro-pillars ranged from 3 μm to 20 μm. (as shown in Figure 2)

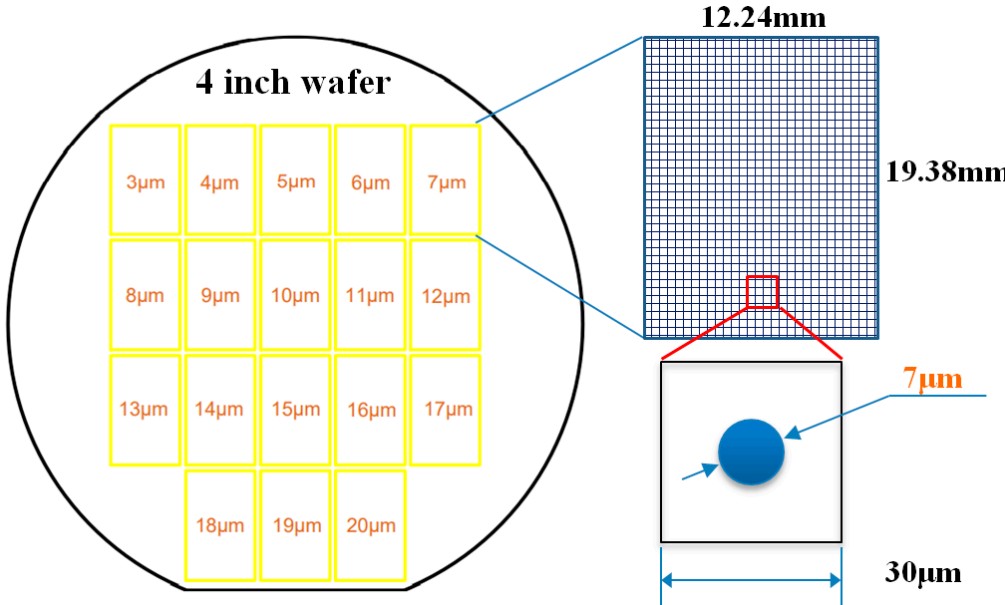

**Figure 2.** Micro-pillar layout on the 4-inch wafer with diameters from 3 to 20 m.

### 2.2. PL and Raman Spectrum Measurements of Laser Confocal Scanning Imaging

After the epitaxy wafer was grown, a room-temperature PL mapping was performed with the Etamax photoluminescence tester. The entire epitaxial wafer was scanned by a 40 mW 325 nm laser beam before and after etching. The PL-mapped image resolution is ~1 mm, a total of 7028-pixels.

Then, the single micro-pillars were characterized by room-temperature Raman spectroscopy. Raman measurements were performed using a Horiba XploRA PLUS confocal Raman microscopy system, taking the Z incident direction along the wurtzite GaN *c*-axis and X direction perpendicular to the *c*-axis. The theoretically allowed X polarization $\overline{Z}$ backscattering geometric configuration $Z(X, X)\overline{Z}$ Raman modes for GaN materials are $A_1$(LO) and $E_2^H$ [13]. The 532 nm Raman laser spot diameter was 0.72 μm and the diffraction grating was set to 2400 grating/mm (high-resolution mode) in order to improve the spectral-spatial resolution.

### 3. Results and Discussion

### 3.1. InGaN/GaN Micro-Pillar Arrays PL Spectral Analysis

Figure 3a is the PL peak wavelength distribution of the entire epitaxial wafer after epitaxial growth, which is recorded as the ORI (original wafer PL). The average wavelength of the PL spectrum is 463.04 nm and the standard deviation is 0.82 nm. It can be seen that the emission wavelengths generated by the epitaxial wafer are not completely consistent after epitaxial growth.

Figure 3b shows the distribution of the PL peak wavelength obtained after micro-pillar ICP etching according to the layout shown in Figure 2, denoted as AFT (after etching PL). The obvious PL area outline proves that the etching was successful in dividing the wafer into the 18 smaller areas. The peak wavelength distribution characteristics inherited from the original wafer can also be seen in Figure 3b.

The wavelength distribution inhomogeneity caused by the original wafer epitaxy affects the extraction of micro-pillar PL information. In order to suppress these influences, we subtracted the AFT PL from the ORI PL and filtered the peak wavelength information out of the micro-LED pillar area. Then, some abnormal values far from the average in the ORI minus AFT PL results caused by etching damage and wafer boundary epitaxy quality degradation are filtered based on the Chauvenet-criterion method [14]. The result is shown in Figure 3c and we can obtain the peak wavelength blueshift caused by the micro-LED pillar arrays. There is a wavelength shift of nearly 0–3 nm in the regions of

different micro-column sizes. It can be concluded that the smaller the column size, the greater the wavelength blueshift.

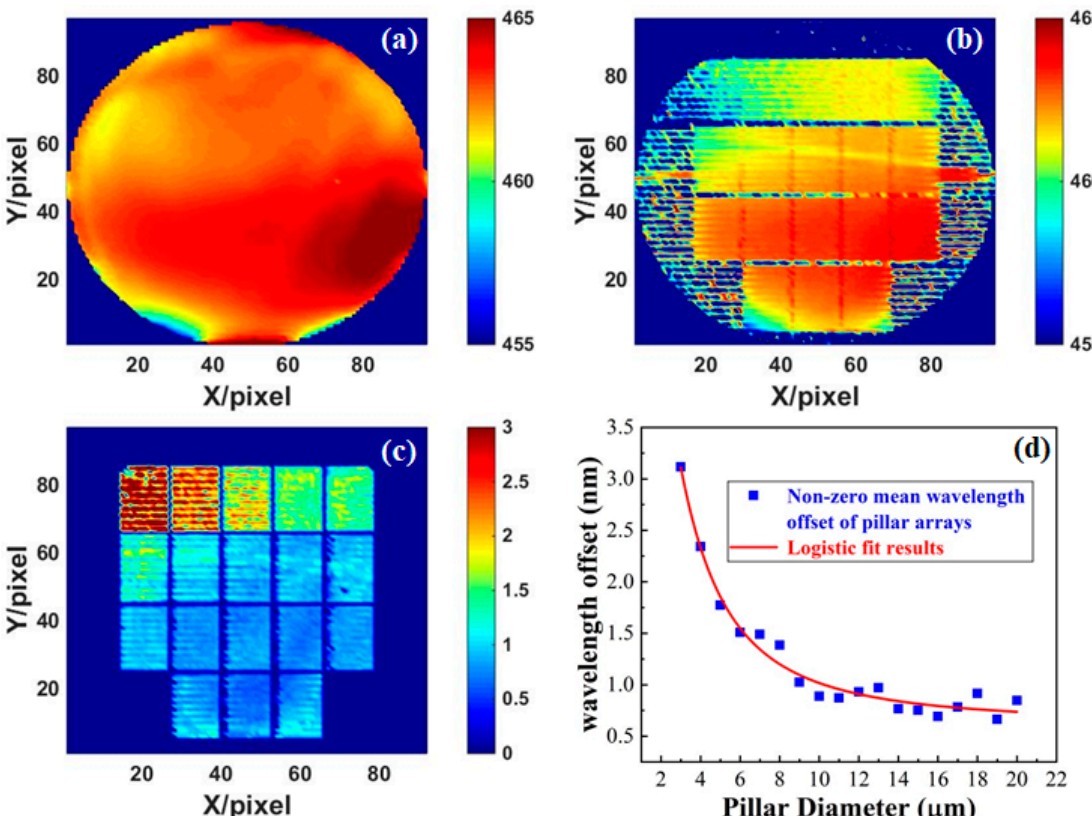

**Figure 3.** Features of PL spectral data: (**a**) PL spectrum of the GaN blue LED epitaxial wafer before etching, denoted as ORI; (**b**) PL spectrum of the epitaxial wafer after micro-pillar etching, denoted as AFT; (**c**) micro-pillar arrays' wavelength shift, ORI-AFT wavelength shift; (**d**) average wavelength shift for different pillar sizes.

To further illustrate micro-pillar arrays' peak wavelength shift, we employed a non-zero mean method to deal with the peak wavelength of each pillar region. As shown in Figure 3d, the wavelength blueshift gradually decreased from 3.1166 nm in the 3 μm pillar size region to a minimum of 0.6634 nm in the 19 μm pillar size region. We attribute these PL peak wavelength blueshifts to the InGaN/GaN multi-quantum well stress release at the side edge of the micro-pillar. As concluded in previous studies [10–12], the InGaN/GaN quantum well sidewalls' strain relaxation leads to the reduction of the piezoelectric polarization field in the quantum well. Affected by this, the equivalent InGaN/GaN band gap becomes larger and the emission wavelength is blueshifted.

### 3.2. Single InGaN/GaN Micro-Pillar Raman Measurements and Analysis

In order to verify the above micro-pillar arrays' strain-related PL results, confocal Raman microscopy measurements were employed to characterize micro-pillar strain relaxation. Figure 4 shows the InGaN/GaN wafer Raman spectra collected in the geometric configuration of $Z(X,X)\overline{Z}$. There are distinct peaks at approximately 567 cm$^{-1}$ and 734 cm$^{-1}$, which are assigned to the $E_2^H$ and $A_1^{LO}$ modes of wurtzite GaN, respectively. The Raman $E_2^H$ mode shift can provide strain relief information due to the biaxial stress caused by lattice mismatch between GaN and InGaN [15,16].

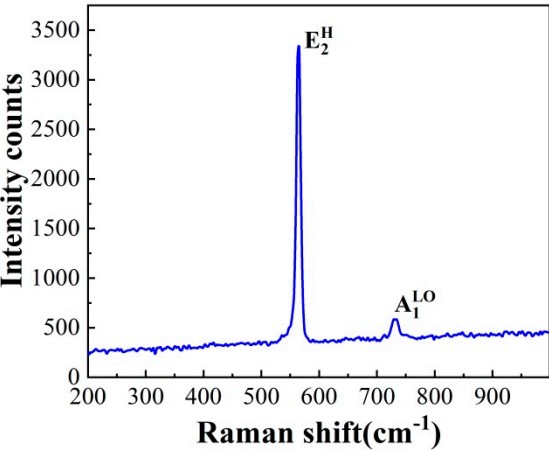

**Figure 4.** The Raman spectra of InGaN/GaN blue LED wafer.

Through our measurements, the strain relaxation at the edge of the micro-pillar has the same characteristics, so we choose a large InGaN/GaN pillar (diameter of 19 μm) to analyze more Raman information on the pillar surface. The center position of the cylindrical sample was taken as the Raman test starting point, the step size along the radius was approximately 1 μm, and the endpoint was the cylinder boundary. Figure 5a shows the $E_2^H$ mode from the center to the edge and it can be seen that the Raman $E_2^H$ mode tends to shift to the right as the distance increases.

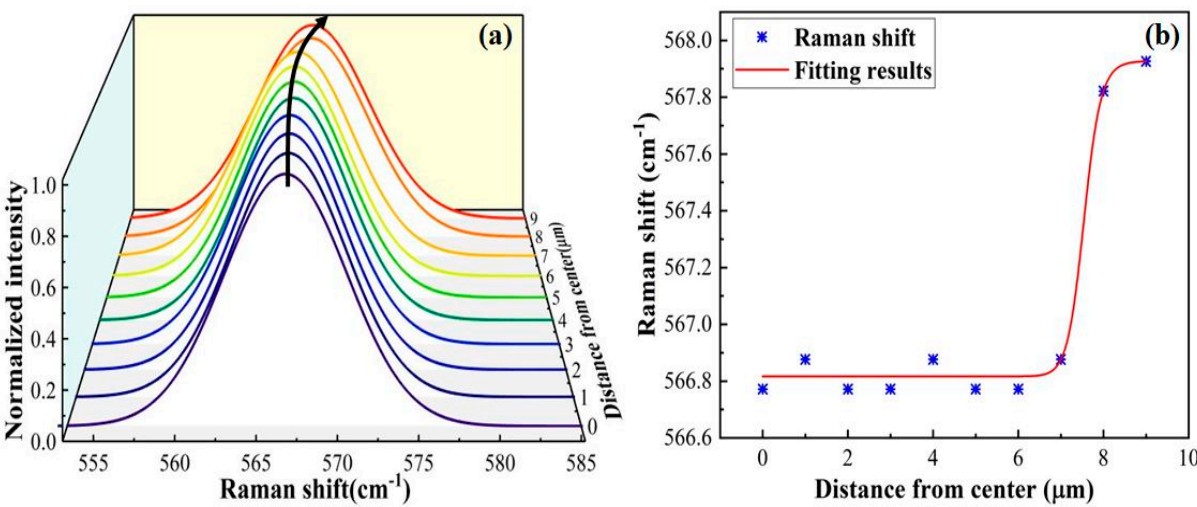

**Figure 5.** (**a**) $E_2^H$ modes of Raman spectra after Gaussian fitting and normalization at different distances from center; (**b**) $E_2^H$ peaks at different distances from the center.

The peak of the $E_2^H$ mode at a different distance from the center was collected in Figure 5b and the movement of the $E_2^H$ peak position could be observed in more detail. As the distance from the center of the micro-pillar increases, the frequency of the Raman peak remains relatively stable near 566.82 cm$^{-1}$ until a distance $\geq$ 8 μm, when the Raman frequency shift starts to move significantly towards the high-frequency direction. At the boundary position, the frequency shift reaches 567.93 cm$^{-1}$, which is close to the bulk GaN $E_2^H$ mode. The change of $E_2^H$ indicates that there is strain inside the pillar, while there is full strain relaxation at the edge.

The stress and strain relief at the pillar edge along the *x*-axis ($\Delta\sigma_{xx}$, $\Delta\varepsilon_{xx}$) can be calculated through Equations (1) and (2).

$$\Delta\sigma_{xx} = \frac{\Delta\omega}{k} = \frac{Edge\ Raman\ shif\left(E_2^H\right) - Center\ Raman\ shift\left(E_2^H\right)}{2.56} \tag{1}$$

$$\Delta\varepsilon_{xx} = \frac{\Delta\sigma_{xx}}{C_{11} + C_{12}} \tag{2}$$

Here, $k$ is the stress coefficient, and $C_{11}$ and $C_{12}$ are the elastic moduli of In$_{0.25}$Ga$_{0.75}$N (listed in Table 1).

From our fitted results in Figure 5b, the Raman frequency shift near the center area is 566.82cm$^{-1}$ and the Raman frequency shift at the edge position is 567.93 cm$^{-1}$. According to Equations (1) and (2), it can be calculated that there is a stress release of $\Delta\sigma_{xx}$ = 0.43 GPa and a strain release of $\Delta\varepsilon_{xx}$ = 0.10% at the pillar edge.

Furthermore, the micro-pillar strain-related bandgap and peak wavelength shift were derived from Equations (3)–(8). The strain analysis parameters are summarized in Table 1.

Considering a strained InGaN-layer wurtzite crystal pseudomorphically grown along the *c*-axis on a thick GaN layer, the strain tensor $\varepsilon$ in the InGaN well region has the following elements [17]:

$$\varepsilon_{xx} = \varepsilon_{yy} = \frac{a_{GaN} - a_{InGaN}}{a_{InGaN}} \tag{3}$$

$$\varepsilon_{zz} = -2\frac{C_{13}}{C_{33}}\varepsilon_{xx} \tag{4}$$

$$\varepsilon_{xy} = \varepsilon_{xz} = \varepsilon_{yz} = 0 \tag{5}$$

where $a_{InGaN}$ and $a_{GaN}$ are the stress-free lattice parameters of the well and barrier layers, respectively. In this paper, the indium composition in the well layer is 0.25. and the unstressed lattice parameters and elastic constants of In$_{0.25}$Ga$_{0.75}$N can be calculated by linear interpolation of the data in [18].

The strain-induced band gap changes were considered, according to the following equations:

$$E_g^{strain} = E_g + (a_{cz} - D_1 - D_3)\varepsilon_{zz} + (a_{ct} - D_2 - D_4)\left(\varepsilon_{xx} + \varepsilon_{yy}\right) \tag{6}$$

where $a_{cz}$ and $a_{ct}$ are the conduction-band deformation potentials along the *c*-axis and perpendicular to the *c*-axis, respectively. $D_1$, $D_2$, $D_3$, and $D_4$ are the shear deformation potentials.

With Equations (4) and (6), the micro-pillar band gap and wavelength shift ($\Delta E_g^{strain}$, $\Delta\lambda$) between the center and edge due to strain relief can be expressed in Equations (7) and (8):

$$\begin{aligned} \Delta E_g^{strain} &= E_g^{strain}(Edge) - E_g^{strain}(Center) \\ &= 2\left(a_{ct} - D_2 - D_4 - \frac{C_{13}}{C_{33}}(a_{cz} - D_1 - D_3)\right)\Delta\varepsilon_{xx} \end{aligned} \tag{7}$$

$$\Delta\lambda = \frac{h \times c}{E_g^{strain}(Edge)} - \frac{h \times c}{E_g^{strain}(Center)} \tag{8}$$

where $c$ is the speed of light in a vacuum and $h$ is Planck's constant.

According to Equations (3)–(8) and our Raman analysis, $\Delta\sigma_{xx}$ and $\Delta\varepsilon_{xx}$, the peak wavelength shift of GaN/InGaN pillar between the center and edge reached 2.45 nm. The Raman shift data show that the strain relaxation at the edge of the pillar is notable, which also provides the reason as to why the small-size InGaN/GaN pillar has a large variation in PL (Figure 3d).

**Table 1.** Parameters used for GaN and InGaN materials stress and strain analysis [18,19].

| Material | InN | GaN | $In_{0.25}Ga_{0.75}N$ |
|---|---|---|---|
| k ($cm^{-1}$/GPa) | | 2.56 | |
| a (Å) | 3.545 | 3.189 | 3.278 |
| $a_{cz}$ (eV) | −3.5 | −4.9 | −4.55 |
| $a_{ct}$ (eV) | −3.5 | −11.3 | −9.35 |
| $D_1$ (eV) | −3.7 | −3.7 | −3.7 |
| $D_2$ (eV) | 4.5 | 4.5 | 4.5 |
| $D_3$ (eV) | 8.2 | 8.2 | 8.2 |
| $D_4$ (eV) | −4.1 | −4.1 | −4.1 |
| $C_{11}$ (Gpa) | 223 | 390 | 348.25 |
| $C_{12}$ (Gpa) | 115 | 145 | 137.5 |
| $C_{13}$ (Gpa) | 92 | 106 | 102.5 |
| $C_{33}$ (Gpa) | 224 | 398 | 354.5 |
| $C_{44}$ (Gpa) | 48 | 105 | 90.75 |

*3.3. InGaN/GaN Quantum Well Finite Element Method (FEM) Simulation*

From the above section, the strain's relaxation was induced by the Raman shift of the GaN layer, we assume that the stress in the GaN layer is equal to that in the InGaN layer, and the direction is opposite. Then, we use the change of the GaN stress to characterize the stress relaxation in the InGaN quantum well but each InGaN quantum well strain cannot be described exactly.

To support the edge strain relaxation and micro size effects described above, a solid mechanic's finite element model in COMSOL Multiphysics was built to simulate the strain distribution in a single micro-pillar. The isotropy of the in-plane strain in the [0001] growth direction and the cylindrical symmetry simplified the simulation to a two-dimensional case [20]. The simulation model was built according to the structure shown in Figure 1, with the sidewall edge of the pillar set as the start of the *x*-axis. The free boundary conditions were assigned to the micro-pillar side walls, and the up and bottom GaN surfaces. Meanwhile, we assumed that the quantum wells are pseudo-morphically grown. The initial strain $\varepsilon_0$ between InGaN and GaN layers, calculated by the lattice mismatch Equation (3), was set as the initial strain boundary conditions [21]. The strain and stress were near the quantum well boundary and the carrier radiative recombination was mainly in the quantum wells so that compressive strain of the InGaN quantum wells was observed.

The FEM calculated strain distribution of 12 $In_{0.25}Ga_{0.75}N$ quantum wells is shown in Figure 6a. The height marks the vertical position of the quantum wells in the micro-LED pillar. At the transverse inside of the quantum well (distance from edge > ~600 nm), the in-plane strain ($\varepsilon_{xx} = \varepsilon_{yy}$) of InGaN is about 2.67%, which is the complete strain of $In_{0.25}Ga_{0.75}N$ on GaN. The strain decreases as the boundary approaches. The strain $\varepsilon_{xx}$ is about 2.3% near the edge of pillar and the edge strain is too small to represent using a color scale. Therefore, the exact strain distribution is listed in Figure 6b.

Figure 6b shows the calculated average strain of the 12 pairs of QW in a 19 μm diameter pillar. The results show that in the micro-pillars, the strain distribution mainly can be divided into three regions. The first region is the complete strain region, which is located in the middle of the micro-pillars and the strain remains constant at 2.67%. The second region is the gradual strain release region, where the strain decreases from 2.67% to 2.36% within ~600 nm to ~10 nm of the pillar edge. The third area is the mutation region, where strain is released rapidly within 10 nm from the boundary and the strain change to 0.25% at the edge of the pillar.

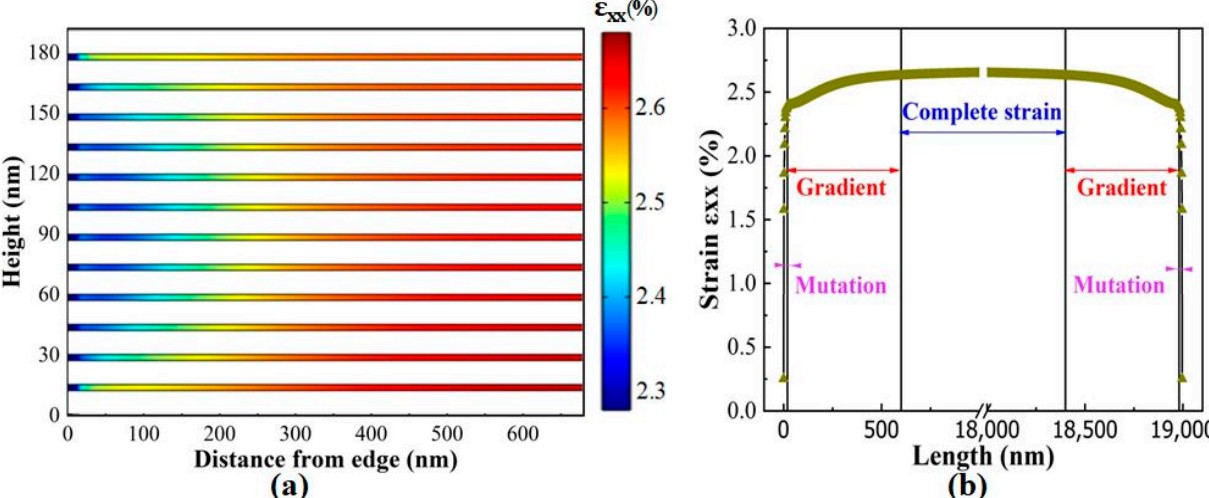

**Figure 6.** The simulated strain ($\varepsilon_{xx}$) of $In_{0.25}Ga_{0.75}N$ well layer with a diameter of 19 μm GaN/InGaN pillar: (**a**) the strain distribution at the boundaries of the 12 quantum wells; (**b**) the division of strained regions after synthesizing 12 pairs of wells.

Furthermore, based on Equations (3)–(8), the quantum well strain-related PL peak emission was estimated and shown in Figure 7a. In the complete strain region, the quantum well peak wavelength is 459 nm. In the strain mutation region, the strain relaxation makes the peak wavelength move to 416 nm. The emission wavelength in the strained relaxation region is blueshifted by up to 43 nm.

According to the cylindrical symmetry, the distance from the edge (in Figure 7a) is assumed to be the micro-LED pillar radius. With the simulation results in Figure 7a, we statistically analyzed the intensity of the peak wavelengths in the 12 quantum wells in the radius of 100 nm, 200 nm, 400 nm, and 800 nm. The results are shown in Figure 7b. According to our simulations, the 800 nm radius pillar emission peak wavelength is about 457 nm. When the radius is reduced to 400 nm, 200 nm, and 100 nm, the corresponding peak wavelengths are 456 nm, 455 nm, and 451 nm. The peak wavelength offset changes from 1–4 nm. We can draw a conclusion that the piezoelectric polarization generated by the strain in the quantum well directly affects the radiation wavelength. The range of strain relaxation and wavelength shift is mainly related to the thickness of the quantum well and the size of the pillar. The smaller the pillar size, the greater the size modulation effect on the peak wavelength of the device.

With this method, the simulations were expanded to a series of columns with diameters from 3 μm to 20 μm and the peak wavelength shift versus pillar size was explored. As shown in Figure 7c, the red datapoints represent the calculated peak wavelength of GaN/InGaN pillars of different diameter. When the column diameter increases from 3 to 20 um, the peak wavelength shift calculated by the simulation decreases from 1.92 nm to 0.32 nm. With an increase in the diameter of the column, the proportion of the strain relief area decreases continuously and the influence on the peak wavelength of the whole micro-column is weakened.

The black datapoints in Figure 7c represent the results of the corresponding PL experiment (also in Figure 3), which are in good agreement with the trends of the FEM simulation results. However, there is a 0.5–1.1 nm difference between the simulation and the PL test data. We attribute these offset deviations to the following reasons: the P GaN on the original wafer is etched to 100 nm, which will create a strain relaxation from the top of the InGaN/GaN quantum well, and subsequently create a peak wavelength offset increase between the original wafer PL (ORI) and the PL wafer after ICP etching (AFT). On the other hand, the GaN micro-pillar over-etching in the direction of the diameter will create extra edge strain relaxation, which is another reason for the enhancement of the PL peak

wavelength offset. These results confirm that the micro size has a modulation effect on the devices' emission wavelength.

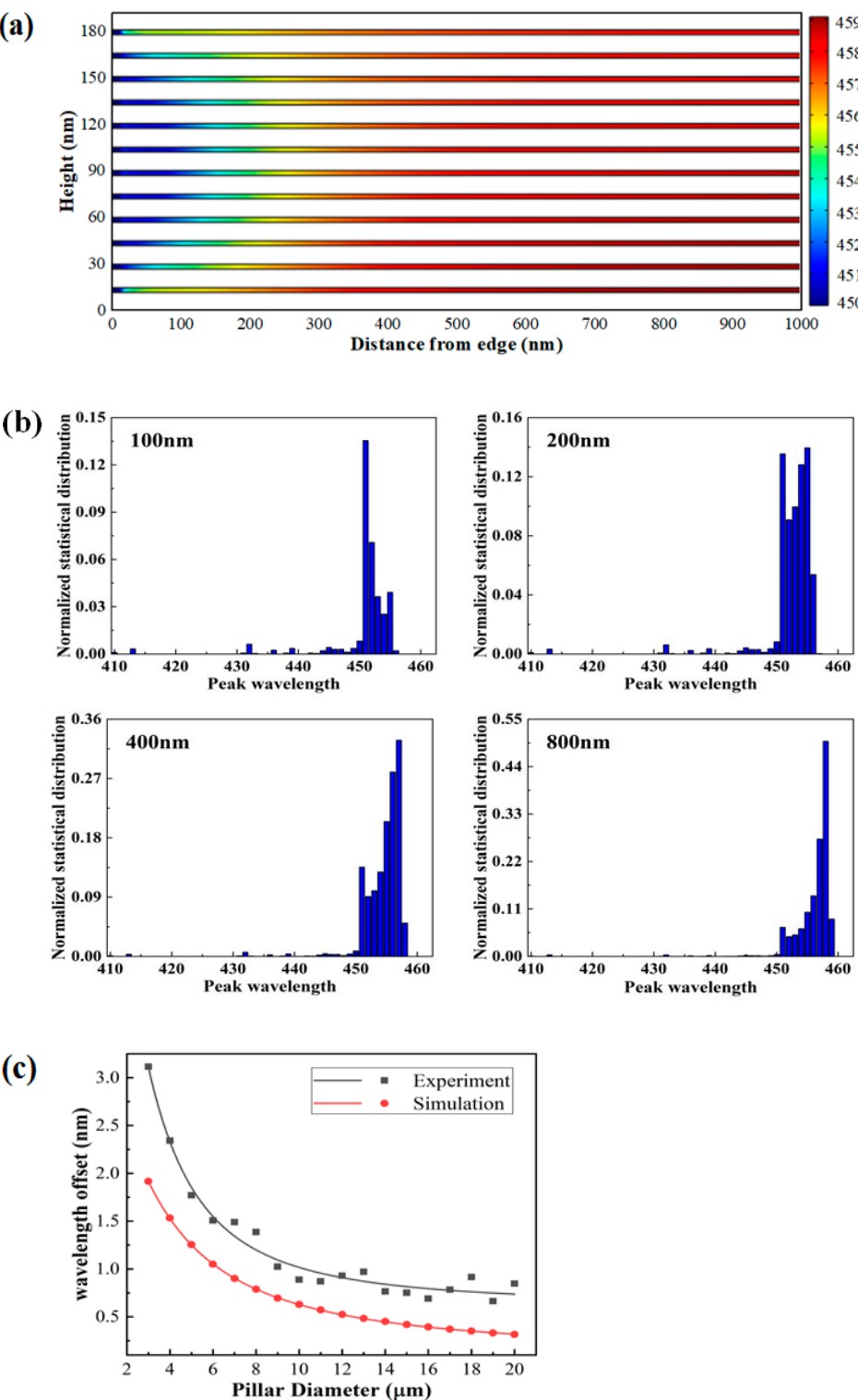

**Figure 7.** (**a**) The simulated peak wavelength distribution (0–1000 nm) of $In_{0.25}Ga_{0.75}N$ well layer with a diameter of 20 μm; (**b**) the intensity of peak wavelengths in 12 pairs of wells in the radius of 100 nm, 200 nm, 400 nm, 800 nm, starting from the boundary; (**c**) experimental and simulated peak wavelength shift as a function of pillar diameter.

## 4. Conclusions

In this work, we prepared a 4-inch GaN blue LED wafer by metal–organic chemical vapor deposition (MOCVD). Then, the wafer was divided into 18 areas and each area was fabricated into micro-LED pillar arrays by ICP etching. The pillar diameter of the 18 areas changes from 3 μm to 20 μm. The relationship between the micro-LED pillars' strain relaxation and peak wavelength shift was researched.

Through our PL mapping measurements, the 3 μm micro-LED pillars peak wavelength blueshifts up to 3.1166 nm due to the InGaN/GaN multi-quantum well strain release at the side edge of the micro-pillar. Then, confocal Raman microscopy measurements were performed, and the existence of the pillar edge strain relaxation was verified by E2H mode Raman shift. Finally, the above results were simulated using the finite element method (FEM) in COMSOL software.

This work shows that with the reduction of micro-LED pillar size, the influence on the peak wavelength shift of micro-LEDs increases. The smaller the device size, the more obvious the strain relaxation effect on device peak wavelength. In the future, the high resolution of micro-LED displays could allow the size of pixels and devices to be less than 10 μm. More attention should be paid to the strain relaxation effect on the LED peak wavelength shift. The results of this paper provide important guidance for the design and application of micro-LED devices.

**Author Contributions:** Conceptualization, H.L. and P.N.; methodology, P.S.; investigation, F.W. and Z.C.; resources, S.L., Y.W. and D.Z.; data curation, K.G.; writing—original draft preparation, C.Z. All authors have read and agreed to the published version of the manuscript.

**Funding:** This research was funded by the Tianjin Municipal Science and Technology Bureau, Grant 18JCYBJC85400, Grant 18ZXCLGX00090, Grant 20JCQNJC00180 and Grant 19JCTPJC48000; in part by the China Scholarship Council (CSC), Grant 201809345004; and in part by the Tianjin Key Laboratory of Optoelectronic Detection Technology and System under Grant TD13-5035 and Grant 2017ZD06.

**Institutional Review Board Statement:** Not applicable.

**Informed Consent Statement:** Not applicable.

**Data Availability Statement:** Not applicable.

**Conflicts of Interest:** The authors declare no conflict of interest.

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
