# Peer review of "Strain Relaxation Effect on the Peak Wavelength of Blue InGaN/GaN Multi-Quantum Well Micro-LEDs"

_applsci, doi:10.3390/app12157431_

Round 1
Author Response
Manuscript applsci-1808881 Response to Reviewer 1:
Comments
- The abstract is a very confusing and suggested to rephrase for better reader understanding.
For example, “The peak wavelength shift caused by edge strain relaxation is eliminated by testing the room temperature PL spectrum before and after arrays etching.” What does authors mean by ‘elimination? Also, ‘PL’ is used without having any reference to its abbreviation.
Authors’ Response:
Thanks for your suggestion, the abstract have been revised carefully and the result is as follows:
Revise: In this paper, the edge strain relaxation of InGaN/GaN MQW micropillars is studied. The micropillar arrays with a diameter of 3 μm ~ 20 μm are prepared on the blue GaN LED wafer by inductively coupled plasma (ICP) etching. The peak wavelength shift caused by edge strain relaxation is tested by the micro-LED pillar arrays room temperature Photoluminescence(PL) spectrum measurements. The results show that there is a nearly 3 nm peak wavelength shift between the micropillar arrays, which is caused by a high range of strain relaxation region in the small size LED pillar. Furthermore, a 19μm micro-LED pillar's Raman spectrum is employed to observe the pillar strain relaxation. It was found that the Raman E2H mode at the micro-LED pillar edge moved to high frequency, which verified an edge strain relaxation =0.10 %. Then, the exact strain and peak wavelength distributions of the InGaN quantum wells were simulated by the finite element method, which provides effective verification of our PL and Raman strain relaxation analysis. The results and methods in this paper have good references for the design and analysis of small-size micro-LED devices.
(see in the Abstract)
Comments
- Manuscript title is misleading. As per the manuscript, there is a shift in the wavelength seen in the fabricated devices and no wavelength modulation.
Authors’ Response:
Thank you for your advice and it has been changed. The revised title as follows on :
Revise: “Strain relaxation effect on emission wavelength of blue InGaN/GaN multi-quantum well micro-LED”
(see in the title)
Comments
- In page 2, what does authors mean by ‘free boundary’.
Authors’ Response:
"Free boundary" means that there are no constraints on the quantum well sidewalls
The result of the change is in line 65 page 2.
Revise: The confocal Raman tests are also performed on the single Micro-LED pillars to demonstrate the release of lattice mismatch stress on the quantum well sidewalls.
(See in in line 65 page 2)
Comments
- Fig 2. is misleading. It’s not clear ‘how can a 7um region can be enlarged to 2.2mm x 19.4mm.
Authors’ Response:
Thank you so much for pointing out our drawing mistakes. Figure 2 has been corrected as follows:
And we have modified the description of Figure 2.
Original : “Then, according to the layout shown in Figure 2, the wafer is etched into 3 μm to 20 μm diameter pillar arrays, the pillar period is 30 μm and the etching depth is 3 μm until the undoped GaN.”
Revise: “Then, according to the layout shown in Figure 2, we divided 18 areas of the same size on a 4-inch wafer, and the area size was 12.24mm×19.38mm. We etched an equal number of micropillars of different sizes in each area, and the etching period was 30 μm (row: 12240 μm /30 μm=408, column: 19380 μm/30 μm=646). The diameters of the etched micropillars range from 3 μm to 20 μm.”
(see in line 76 to 81, page 2)
Comments
- In Page 3, its little unclear what does authors mean by ‘The PL-mapped pixel size is ~1mm, a total of 7028 groups’. Also, what does mean? It needs to be specified clearly in manuscript. Again, in the same page, Abbreviation for ORI, AFT are missing.
Authors’ Response:
~1mm refers to the resolution of PL test. Then, after a full scan of the 4-inch wafer, a total of 7028 sets of data were obtained.
Revise:……. The PL-mapped image resolution is ~1mm, a total of 7028-pixels. (see in line 89 to 90, page 3)
The “” is used to describe the Raman scattering geometry: outside the bracket, the symbols show from left to right the direction of incident and scattered light, respectively, and inside the bracket, they give from left to right the polarization direction of the incident and scattered light, respectively. Which is explained in ref [13].
Revise:
Raman measurements were performed using a Horiba XploRA PLUS confocal Raman microscopy system, where taking Z incident direction along the wurtzite GaN c-axis and X direction perpendicular the c-axis. An X polarization ZÌ… backscattering geometric configuration Z(X, X)ZÌ… theoretically allowed Raman modes for GaN materials are A1(LO) and E2H.
(See in line 92 to 96, page 3)
Abbreviation for ORI, AFT are illustrated in line 102 and line 107, Page 3
Comments
6.Again, in page 3, authors mention, “The peak wavelength distribution inhomogeneity matching Figure 3(a) can also be observed”. How? And a detailed description is missing. Also, in this paragraph what does ‘abnormal value’ mean and how the conclusion is made? such details are missing.
Authors’ Response:
Figures 3(a) and 3(b) are the epitaxial wafers before and after etching, respectively. They are all epitaxially grown under the same growth conditions, so they have uniform wavelength distribution inhomogeneity.
Revise: The peak wavelength distribution characteristics inherited from the original wafer can also be seen in the Figure3 (b).
(see in line 109 and 110, page 3)
Data that are far from the mean due to etch damage and interference from boundary fixtures during testing are referred to as outliers. Outlier determination is based on the “Chauvenet Criterion”.
Revise: Then, some abnormal values far from the average in the ORI minus AFT PL results, which is caused by etching damage and wafer boundary epitaxy quality degradation, are filtered based on the Chauvenet-criterion method [14].
(see in line 114 and 116, page 3)
Comments
- In section 3.3, is the model build for a single pillar or for multiple pillars? this is not clear in the description. Also, displaying the modelled geometry with the appropriate description is helpful for better reader understanding.
Thanks for your opinion. We have made changes as follows.
Revise: To support the edge strain relaxation and micro size effects described above, the solid mechanic’s finite element model in COMSOL Multi-physics was built to simulate the strain distribution in a single micro-pillar.
(See in line 206, page 8)
Comments
- In Fig 6 (a), what does it mean by height? Also, it’s unclear what does the color scale mean? It needs to be specified. Here, authors mention that “The strain decreases as the boundary approaches. The strain εxx is 0.25% at the edge of the pillar.” But from the Fig 6(a), its shows ~2.3%. Its unclear how the number 0.25% has come from? Fig 6(a) and 6(b) doesn’t co-relate well as the axis are different. Mainly, in the presented surface plot these regions are not identifiable.
Authors’ Response:
Thank you for your suggestions.
The height corresponds to the positions of the well layer and the barrier layer during quantum well epitaxy growth (starting from the bottom of the lowest GaN layer as 0).The color scale εxx (%) in Figure 6(a) has been added. The modified Figure 6 is shown below.
Figure 6
About the Fig 6 (a) and (b) illustration, we revise it as follows:
The FEM calculated strain distribution of 12 In0.25Ga0.75N quantum wells is shown in Figure 6(a), the height marks the vertical position of the quantum wells in the micro-LED pillar. At the transverse inside of the quantum well (Distance from edge>~ 600 nm), the in-plane strain (εxx=εyy) of InGaN is about 2.67 %, which is the complete strain of In0.25Ga0.75N on GaN. The strain decreases as the boundary approaches. The strain εxx is about 2.3% near the edge of pillar and the edge strain is too small to represent by color scale. Therefore, the exact strain distribution is listed in the Figure 6 (b).
(see from line 221 to 227, page 8)
Comments
- Missing units in Fig 7 description, “Starting from the boundary, in the range of 0-100, 0-200, 0-400, 0-800, the intensity of peak wavelengths in 12 pairs of wells”.
Authors’ Response:
Thanks for your suggestion, the unit has been added in Figure 7 and the illustration of the radius has been add in line 245~248.
Revise:
According to the cylindrical symmetry, the distance from the edge (in Fig 7.a) is assumed to be the micro-LED pillar radius. With the simulation results in Fig. 7(a), we statistics the intensity of the peak wavelengths in the 12 quantum wells in the radius of 100nm, 200nm, 400 nm, and 800nm.
(see from line 248 to 251 page 10)
Comments
- In page 9, authors mention, “Furthermore, based on equation (3~8), the quantum well strainrelated PL peak emission was estimated and shown in Fig. 7(a)”. Does this mean this graphic is plotted using theoretical formulation?
Authors’ Response:
Yes, the Fig. 7 distribution of peak wavelengths is calculated based on (Fig. 6 a) the data of the strain distribution.
Comments
- Again, in page 9, its unclear what does authors mean by “According to our simulation, the 800 nm radius pillar emission peak wavelength is about 457 nm, and the peak wavelength blue shifts to 456nm, 455nm, and 451nm, as the pillar radius, reduces to 800 nm, 240 400nm, 200nm, 100nm”. Is this basing on the Fig 7(c)?
Authors’ Response:
Thanks for your suggestion, it is based on Figure 7(b), we changed the expression, and the result of the change is as follows:
Revise:
The results are shown in Figure 7(b). According to our simulations, the emission peak wavelength of a column with a radius of 800 nm is about 457 nm. When the radius is reduced to 400 nm, 200 nm and 100 nm, the corresponding peak wavelengths are 456 nm, 455 nm and 451 nm.
(see from line 251 to line 255, page 10)
Comments
- Page 9, authors mention ‘modulation effect’, but it is the shift in the wavelength.
Authors’ Response:
Yes, we agree with you.
In our opinion, the micro-LED size related strain relaxation can modulate the peak radiation of wavelength. So, we use “size modulation effect” on the micro-LED peak wavelength shift.
(see in line 258 and 277, page 10)
Comments
- Conclusion is very confusing and suggested to rephrase for better reader understanding.
Authors’ Response:
The conclusion has been revised as follows:
In this work, we prepared a 4-inch GaN blue LED wafer by metal-organic chemical vapor deposition (MOCVD). Then, the wafer was divided into 18 areas and each area was fabricated into micro-LED pillar arrays by ICP etching. The pillar diameter of the 18 areas changes from 3 μm to 20 μm. The relationship between micro-LED pillars strain relaxation and peak wavelength shift was researched.
Through our PL mapping measurements, the 3 μm micro-LED pillars peak wavelength blue shifts up to 3.1166 nm due to the InGaN/GaN multi-quantum well strain release at the side edge of the micropillar. Then, confocal Raman microscopy measurements were performed and the existence of the pillar edge strain relaxation was verified by the E2H mode Raman shift. Finally, the above results were simulated using the finite element method (FEM) in the COMSOL software.
This work shows that with the reduction of micro-LED pillar size, the influence on the peak wavelength shift of micro-LED increases. The smaller the device size, the more obvious the strain relaxation effect on device peak wavelength. In the future, the high resolution of micro-LED display makes the size of pixels and devices less than 10 μm. More attention should be paid to the strain relaxation effect on the LED peak wavelength shift. The results of this paper provide important guidance for the design and application of micro-LED devices.
Comments
- Overall, the text in the whole manuscript in some sense is unclear. It is obscure for the general audience in term of both language and scientific understanding. The explanation of the findings in some sense misleading and need modification.
Authors’ Response:
Thanks for the reviewers’ suggestions, the paper language and scientific explanation has been revised.
Image upload failed in Response.
Please see attached manuscript for details of changes.

Reviewer 2 Report
Title: Strain relaxation effect on InGaN/GaN Multi Quantum Wells (MQWS) wavelength modulation of GaN blue Micro-LED
Authors: Chaoqiang Zhang, Ke Gao, Shuai Dong, Xiaoyu Feng, Yuan Meng, Fei Wang, Zhiming Chen, Philip Shields, Sean Lee, Yanqin Wang, Teling Hsia, Boqi Zhan, Dongyan Zhang, Hongwei Liu, and Pingjuan Niu
The manuscript by Chaoqiang Zhang et al. containing new data on the edge strain relaxation of InGaN/GaN micropillars is very timely. International research activity has started for the development of blue LEDs based on the InGaN/GaN MQW structures on a sapphire substrate. One of the crucial problems of this technology is finding the optimum conditions for the device emission. In this connection, the paper is interesting from the practical point of view.
In my opinion, this well-done paper can be recommended for publication in Journal of Applied Sciences.
There are only two issues associated with the manuscript:
1. Have the authors measured the PL spectra at low temperatures? It would be useful to investigate such more informative spectra and make a comparison with the effects at room temperature.
2. There are some inaccuracies in the paper:
2.1 Keywords: “…InGaN/GaN Multi Quantum Wells (MQWS)”;
page 1, lines 35-36: “…a multiple quanta well (MQW) structure”;
but page 1, lines 40-41: “…the InGaN/GaN multiple quantum well (MQW)”.
2.2 Figure 3 a-d: The letters a, b, c aren’t indicated in the corresponding figures.
2.3 Page 6, line 175:
“…the In composition of the well layer is 0.25” > the In content in the well layer is 0.25.
2.4 Page 6, line 183-184: “… the micropillar band gap and wavelength change (ΔE???????, Δλ) between the center and edge…”;
page 6, line 186-187: “…Δλ is the wavelength shift”;
page 6, line 188-189: “…the peak wavelength shift …between edge and middle position”.
Author Response
Manuscript applsci-1808881 Response to Reviewer 2:
Comments
- Have the authors measured the PL spectra at low temperatures? It would be useful to investigate such more informative spectra and make a comparison with the effects at room temperature.
Authors’ Response:
Yes, I agree with you very much, the low temperature PL spectroscopic test can get more detailed quantum well information, but our sample is a 4-inch epitaxial wafer, because the whole epitaxial wafer is too large to fit into the low temperature PL sample chamber, So no low temperature PL test was performed. And because there are other uses, we have not cut it at present, but we will conduct low-temperature PL tests in the future to study the more detailed quantum well luminescence mechanism. thank you very much for your suggestion.
Comments
2.1 Keywords: “…InGaN/GaN Multi Quantum Wells (MQWS)”;
page 1, lines 35-36: “…a multiple quanta well (MQW) structure”;
but page 1, lines 40-41: “…the InGaN/GaN multiple quantum well (MQW)”.
Authors’ Response:
Thank you for your advice and it has been changed.The revised text reads as follows on :
Keywords: “…InGaN/GaN Multiple Quantum Well (MQW)”;
page 1, lines 35-36: “…a multiple quanta well (MQW) structure”;
but page 1, lines 40-41: “…the InGaN/GaN multiple quantum well (MQW)”
Comments
2.2 Figure 3 a-d: The letters a, b, c aren’t indicated in the corresponding figures.
Authors’ Response:
Thank you for pointing this out. The black icons (a), (b), (c) we used originally were not affected by the blue background, so I have changed them to white.
The modified effect is as follows:
Comments
2.3 Page 6, line 175:
“…the In composition of the well layer is 0.25” > the In content in the well layer is 0.25.
Authors’ Response:
Thank you for your advice and it has been changed.
Comments
2.4 Page 6, line 183-184: “…the micropillar band gap and wavelength change (Δ E???????, Δλ) between the center and edge…”;
page 6, line 186-187: “…Δλ is the wavelength shift”;
page 6, line 188-189: “…the peak wavelength shift …between edge and middle position”.
Authors’ Response:
Thank you for your advice and it has been changed the interpretation of Δλ has been unified as “wavelength shift… between the center and edge”
Please refer to the attachment for the revised paper

Reviewer 3 Report
The authors present an interesting study on the effect of strain relaxation on emission wavelength in InGaN/GaN multi-quantum well micro-LEDs and its dependence on LED diameter. The study combines PL and Raman measurements, and the results are supported by finite element strain simulations.
Generally, the paper is well written and easy to read.
I suggest some minor points to be addressed, before publication:
1) I suggest to change the title into "Strain relaxation effect on emission wavelength of blue InGaN/GaN multi-quantum well micro-LEDs", or at least to eliminate the acronym MQWS and the "of GaN"
2) I suggest to change "edge strain relaxtion" into "strain relaxtion at the surface of"
3) I suggest to write out ICP at its first appearance.
4) There are a few misspellings or wrong words:
abstract, line 13, the term "eliminated" probably should be "determined" or similar
abstract, line 22 the meaning of "Spectral Research" is not clear
introduction, line 35: quantum instead of quanta
page 9, line 236: "We statistics the intensity..."
5) on page 3, the Chauvenet-criterion maybe would need a reference
6) In Fig. 6b, the text labels are a bit too small
7) I suggest to give more details on the simulation, in section 3.3. For example, has cylindrical symmetry been used, how exactly has the initial strain been included? Were free boundary conditions applied only to sidewalls or also to top and/or bottom? Also, is there a reason why in Fig. 6a strain is shown only in the QWs? Towards the surface there should be some tensile strain in the barriers. Maybe it would be more consistent to show strain including its sign?
8) The discussion of Fig. 7b is not totally clear. I guess it is the experimental data. Is there a reason for the oscillatory behaviour of the experimental data in Fig. 7c?
Author Response
Manuscript applsci-1808881 Response to Reviewer 3:
Comments
1) I suggest to change the title into "Strain relaxation effect on emission wavelength of blue InGaN/GaN multi-quantum well micro-LEDs", or at least to eliminate the acronym MQWS and the "of GaN"
Authors’ Response:
Yes, the title you suggested is a better option, I've made changes.
Comments
2) I suggest to change "edge strain relaxtion" into "strain relaxtion at the surface of"
Authors’ Response:
Thank you for your advice but the strain relief does not only exist on the surface of each layer, and the strain relief is mainly caused by the unconstrained sidewall boundary, so we prefer the edge strain relaxation.
Comments
3) I suggest to write out ICP at its first appearance.
Authors’ Response:
Thanks for the suggestion, the first occurrence of the ICP is on line 13 of the abstract, it has been changed as requested.
Comments
4) There are a few misspellings or wrong words:
abstract, line 13, the term "eliminated" probably should be "determined" or similar
abstract, line 22 the meaning of "Spectral Research" is not clear
introduction, line 35: quantum instead of quanta
page 9, line 236: "We statistics the intensity..."
Authors’ Response:
Thank you, all above misspellings have been changed.
Comments
5) on page 3, the Chauvenet-criterion maybe would need a reference
Authors’ Response:
Yes, thanks for the suggestion and references have been inserted.
Comments
6) In Fig. 6b, the text labels are a bit too small
Authors’ Response:
Thanks for your suggestion and the text labels have been changed in Figure 6(b).
The modification result is as follows:
Comments
7) I suggest to give more details on the simulation, in section 3.3. For example, has cylindrical symmetry been used, how exactly has the initial strain been included? Were free boundary conditions applied only to sidewalls or also to top and/or bottom? Also, is there a reason why in Fig. 6a strain is shown only in the QWs? Towards the surface there should be some tensile strain in the barriers. Maybe it would be more consistent to show strain including its sign?
Authors’ Response:
Thank you, following your suggestions, more simulation details have been added in section 3.3 from line 206 to line 215.
About the strain sign, the InGaN layer affected by compressive strain and the default sign is “+”, we add some words at line 215 as our response.
The revise result as follows:
To support the edge strain relaxation and micro size effects described above, the solid mechanic’s finite element model in COMSOL Multi-physics was built to simulate the strain distribution in the micro-pillars. The isotropy of the in-plane strain in the [0001] growth direction and the cylindrical symmetry simplifies the simulation to a two-dimensional case[20]. The simulation model is built according to the structure shown in Figure 1, the sidewall edge of the pillar is set as the start of the x-axis. And the free boundary conditions are assigned in Micro pillar side walls, up and bottom GaN surface. Meanwhile, we assume that the quantum wells are pseudo-morphically grown. The initial strain ε0 between InGaN and GaN layers calculated by the lattice mismatch equation (3) is setting as the initial strain boundary conditions [21]. The strain is near the quantum well boundary and the carrier radiative recombination is mainly in quantum wells, so the compressive strain of InGaN quantum wells were observed.
Comments
8) The discussion of Fig. 7b is not totally clear. I guess it is the experimental data. Is there a reason for the oscillatory behavior of the experimental data in Fig. 7c?
Authors’ Response:
Figure 7(b) is the simulation data. We calculated the emission wavelength statistics of the simulation data in the range of 0-100, 0-200, 0-400, 0-800. The purpose is to verify the influence of size on the peak wavelength of the device. In order to explain more accurately and clearly,we revised the words from line 238 to 241, and the legend of Fig 7 (b) was also made a responding change.
The oscillations of the experimental data in Figure 7(c) are caused by wafer epitaxy and data processing. The original LED wafer outside position has a large peak wavelength fluctuation than the center (see in Figure 3.a). While the pillar array in the size regions of 8, 13, 18, 19, 20,17, 12, and 7 (see in Figure 2) are in the outside of the wafer, the data processing method in this paper cannot eliminate wafer peak wavelength fluctuation effect completely. So, the data point in Fig 7 (c) of 7, 8,13, 17, 18, 20 are generally higher than their neighbors.
Although it looks like an oscillatory behavior, which is cause by the random wafer sample characteristics and the repeated array distribution. There is no scientific reason to explain it, so we did not modify this part.
Please see attachment for revised manuscript
